# Constructing Semantic Hierarchies via Fusion Learning Architecture

## Abstract

Semantic hierarchies construction means to build structure of concepts linked by hypernym-hyponym ("is-a") relations. A major challenge for this task is the automatic discovery of hypernym-hyponym ("is-a") relations. We propose a fusion learning architecture based on word embeddings for constructing semantic hierarchies, composed of discriminative generative fusion architecture and a very simple lexical structure rule for assisting, getting an F1-score of 74.20% with 91.60% precision-value, outperforming the state-of-the-art methods on a manually labeled test dataset. Subsequently, combining our method with manually-built hierarchies can further improve F1-score to 82.01%. Besides, the fusion learning architecture is language-independent.

## 1 Introduction

Ontologies and semantic thesauri (Miller, 1995; Suchanek et al., 2007) are significant for many natural language processing applications. The main components of ontologies and semantic thesauri are semantic hierarchies (see in Figure 1). In the WordNet, semantic hierarchies are organized in the form of "is-a" relations. For instance, the words "dog" and "canine" have such relation, and we call "canine" is a hypernym of "dog". Conversely, "dog" is a hyponym of "canine". The hypernym-hyponym ("is-a") relation is the main relationship in semantic hierarchies. However, such manual semantic hierarchies construction as WordNet (Miller, 1995) and YAGO (Suchanek et al., 2007), the primary problem is the tradeoff between coverage scope and human labor. A number of papers have proposed some approach to extract semantic hierarchies automatically.

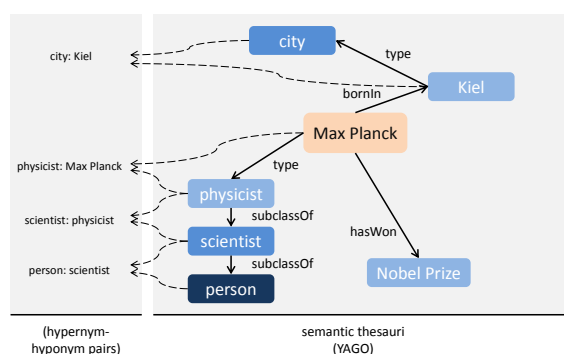

Figure 1: Semantic hierarchies in semantic thesauri

hypernym-hyponym relation discovery is the key point of semantic hierarchies construction, also the major challenge. The usage of the context is a bottleneck in improving performance of hypernym-hyponym relation discovery. Several works focus on designing or learning lexical patterns (Hearst, 1992; Snow et al., 2004) via observing context of hypernym-hyponym relation, which suffer from covering a small proportion of complex linguistic circumstances. Besides, distributional inclusion hypothesis, which states that hypernyms tend to occur in a superset of contexts in which their hyponyms are found. In other words, hypernyms are semantically broader terms than their hyponyms (Kotlerman et al., 2010; Lenci and Benotto, 2012). However, it is not always rational. To acquire more contexts of words, Fu (2013) applies a web mining method to discover the hypernyms of Chinese entities from multiple sources, assuming that the hypernyms of an entity co-occur with it frequently. The method works well for named entities. But for class names with wider range of meanings, this assumption may fail.

Word embedding is a kind of low-dimensional

and dense real-valued vector encoding context information. Inspired by Mikolov (2013b) who founded that word embeddings can capture a considerable amount of syntactic/semantic relations, and found that hypernym-hyponym relations are complicated and a uniform linear projection cannot fit all of the hypernym-hyponym word pairs, Fu (2014) proposed an architecture for learning semantic hierarchies via word embeddings with clustered hypernym-hyponym relation word pairs in advanced. But the method just focuses on linear transformation of word embeddings, using shallow level semantic of the representation. Besides, the method need clustering for hypernym-hyponym relation word pairs in advanced.

Since word embeddings can capture a considerable amount of syntactic/semantic relations (Mikolov et al., 2013b), we considered constructing a uniform architecture for semantic hierarchies learning based on nonlinear transformation of word embeddings. Inspired by advantages of discriminative model and generative model (see in Section 3), we fuse the two kind of models into one architecture. Considering word embeddings encode context information but ignore the lexical structures which contain some degree of semantic information, we integrate a very simple lexical structure rule into the previous fusion architecture aiming at building semantic hierarchies construction (see in Section 3.4).

For evaluation, the experimental results show that our method achieves an F1-score of 74.20% which outperforms the previous state-of-the-art methods. Moreover, and gets a much higher precision-value of 91.60%. Combining our method with the manually-built hierarchy can further improve F-score to 82.01%. The main contributions of our work are as follows:

- We present a uniform fusion architecture which can learn semantic hierarchies via word embeddings without any background knowledge.

- The method we proposed outperforms the state-of-the-art methods on a manually labeled test dataset especially with a good enough precision-value for application.

- The fusion learning architecture is language-independent which can be easily expanded to be suitable for other languages.

## 2   Related Work

During the early phase of semantic hierarchies study, some focused on building manually-built semantic resources, WordNet (Miller, 1995) is a representative thesauri among them. Such manually-built hierarchies have exact structure and high accuracy, but their coverage is limited, especially for fine-grained concepts and entities. Some researchers presented automatic approaches for supplementing manually-built semantic resources. Suchanek et al. (2007) linked the categories in Wikipedia onto WordNet in construction of YA-GO. However, the coverage is still limited by the scope of Wikipedia.

The major challenge for building semantic hierarchies is the discovery of hypernym-hyponym relations automatically. The usage of the context is a bottleneck in improving performance of discovery of hypernym-hyponym relations. Some researchers proposed method based on lexical pattern abstracted from context manually or automatically to mine hypernym-hyponym relations. Hearst (1992) pointed out that certain lexical constructions linking two different noun phases (NPs) often imply hypernym-hyponym relation. A representative example is "such NP1 as NP2". Considering time-consuming of manually-built lexical patterns, Snow et al. (2004) proposed a automatic method extracting large numbers of lexico-syntactic patterns to detect hypernym relations from a large newswire corpus. But the method suffers from semantic drift of auto-extracted patterns. Generally speaking, these pattern-based methods often suffer from low recall-value or precision-value because of the coverage and quality of extracted patterns.

Some measures rely on the assumption that hypernyms are semantically broader terms than their hyponyms. The assumption is a variation of the Distributional Inclusion Hypothesis (Geffet and Dagan, 2005; Zhitomirsky-Geffet and Dagan, 2009). The pioneer work by Kotlerman et al. (2010) designed a directional distributional measure to infer hypernym–hyponym relations. Differently from Kotlerman et al. (2010), Lenci and Benotto (2012) focus on applying directional, asymmetric similarity measures to identify hypernyms. However the broader semantics hypothesis may not always infer broader contexts (Fu et al., 2014).

Considering the behavior of a person explor-

ing the meaning of an unknown entity, Fu (2013) applies a web mining method to discover the hypernyms of Chinese entities from multiple sources. The assumption is that the hypernyms will co-occur with its hyponym entity frequently. But the assumption maybe failed when involved with concept words which have boarder semantic compared with entities. Inspired by the fact (Mikolov et al., 2013b) that word embeddings can capture a considerable amount of syntactic/semantic relations(e.g. $v(\text{king})$ $-$ $v(\text{queen})$ $\approx$ $v(\text{man})$ $-$ $v(\text{woman})$, where $v(w)$ is the word embedding of the word $w$), Fu (2014) present an approach to learn semantic hierarchies with clustered hypernym-hyponym relation word embedding pairs. However the method just focuses on linear transformation of word embeddings, using shallow level semantic of the representation. Besides, the method need clustering for hypernym-hyponym relation word pairs in advanced and the precision-value on test data is not good enough for practical application. Shwartz et al. (2016) included additional linguistic information for LSTM-based learning, but the method has co-occurrence requirements for hyponym-hypernym pairs in corpus.

Enlightened from good properties of word embeddings for capturing semantic relationship between words in work of Fu (2014), we further explore capacity of word embedding for semantic hierarchies using neural networks based on a fusion learning architecture. Above all, the method we proposed do not need clustering for hypernym-hyponym relation word pairs in advanced.

## 3 Method

Given the hypernyms list of a word, our goal is building a semantic hierarchies construction of these hypernyms and the given word(Fu et al., 2014), the process is presented in Figure 2.

The whole information all we have to figure out whether there exists a hypernym-hyponym relation for the word pair is word embeddings representation. Two major kind of architectures for such problem in general, one is discriminative and another is generative.

**Discriminative Architecture:** Discriminative architecture regards the discovery of hypernym-hypernym relation as a classification task, the process of learning semantic hierarchies equal to classify word pair into `yes` or `no` for whether exists

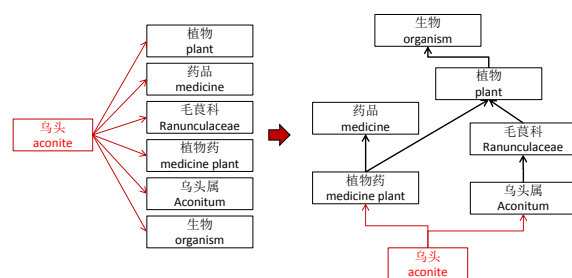

Figure 2: An example of learning semantic hierarchies

hypernym-hypernym relation.

**Generative Architecture:** Generative architecture focus on generating the hypernym of a given hyponym directly. Direct generation is usual impractical, so generative method produces a fake target which is very similar with the true one.

We consider fusing these two models to discovery hypernym-hyponym relation much more precisely. We use Multilayer Perceptron (MLP) (Rosenblatt, 1961) to achieve a generative model and Recurrent Neural Network (RNN) (Mikolov et al., 2010) to implement a discriminative model.

### 3.1 Word Embedding Training

Various methods for learning word embeddings have been proposed in the recent years, such as neural net language models (Bengio et al., 2003; Mnih and Hinton, 2009; Mikolov et al., 2013b) and spectral models (Dhillon et al., 2011). More recently, Mikolov et al. (2013a) propose two log-linear models, namely the Skip-gram and CBOW model, for inducing word embeddings efficiently on a large-scale corpus because of their low time complexity. Additionally, their experiment results have shown that the Skip-gram model performs best in identifying semantic relationship among words. For this reason, we employ the Skip-gram model for estimating word embeddings in this study.

### 3.2 Generative Architecture Based on MLP

Multilayer Perceptron (MLP) (Rosenblatt, 1961) is a feedforward artificial neural network which maps inputs onto a set of appropriate outputs. An MLP consists of multiple layers connecting each layer fully connected to the next one. Except for the input nodes, each node is a neuron with a non-linear activation function(e.g. sigmoid). MLP is a modification of the standard linear perceptron and

can distinguish data that are not linearly separable (Cybenko, 1989).

For single hidden layer MLP, there are input layer $x$, hidden layer $h$ and output layer $y$ range from bottom to top. The value of neurons in each layer is a nonlinear projection of the previous layer. Formalization denotation in formulas are as follows:

$$h = f_h(W_h x + b_h) \qquad (1)$$

$$y = f_y(W_y h + b_y) \qquad (2)$$

Where $W_h, W_y$ are matrices for linear projection. And $f_h, f_y$ are nonlinear activation functions for nonlinear transformation.

In our work, we use Multilayer Perceptron as the main component for generative architecture. The inputs of MLP is word embedding representation of hyponym and outputs a fake hypernym embedding which is very similar with the true hypernym vector. The model produces final result by calculating the distance between the fake hypernym and the candidate hypernym word in continuous space, subsequently, comparing the distance and a predefined threshold to give a judgment. By adjusting the threshold value of similarity, we expect MLP model obtain much higher precision compared with the discriminative one.

### 3.3 Discriminative Architecture Based on RNN

Recurrent Neural Network (RNN) (Mikolov et al., 2010) is a kind of artificial neural network in which connections between neuron form a directed cycle, creating an internal state of the network which allows it to exhibit dynamic temporal behavior according to the history information. Unlike feedforward neural networks (e.g. MLP), RNNs can use their internal memory to process future inputs. The features of RNN makes them applicable to tasks such as unsegmented connected handwriting or speech recognition. There are some variants of original RNN, the most representative one of them is Long Short Term Memory (L-STM) (Hochreiter and Schmidhuber, 1997) which is capable of learning long-term dependencies. In this paper we use the original simple recurrent networks (SRN). There are two major classes of SRN, know as Elman networks (Elman, 1990) and Jordan networks (Jordan, 1997). In this paper, we use Elman networks as the main component for discriminative architecture (see in Figure 3).

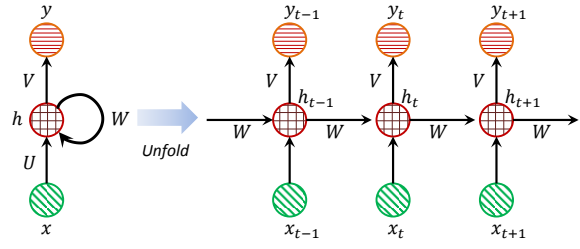

Figure 3: Recurrent Neural Network (Elman networks)

In Elman networks, the main architecture is composed of three classes layers, namely, input layer $x$, hidden layer $h$, and output layer $y$. Different from feedforward neural networks, the hidden layer $h_t$ is depended on the previous time step hidden layer $h_{t-1}$ and the current time step input layer $x_{t-1}$. And the output layer $y_t$ is update by the current time step hidden layer $h_t$. Formalization denotation in formulas are as follows:

$$h_t = f_h(U x_t + W h_{t-1} + b_h) \qquad (3)$$

$$y_t = f_y(V h_t + b_y) \qquad (4)$$

Where $U, V, W$ are matrices for linear projection. And $f_h, f_y$ are nonlinear activation functions for nonlinear transformation.

The inputs of RNN is word embeddings representation of hyponym and candidate hypernym sequence. We regard the hyponym and candidate hypernym as a sequence, because that the judgment of candidate hypernym is depended on hyponym in discrimination process. Ignoring the outputs during recurrent process, we take the final output of the last input in the sequence as the result of discrimination.

### 3.4 Fusion Learning Architecture Combined with a Simple Lexical Structure Rule

Generative architecture can get a very high precision by adjusting the threshold value of similarity, but will pay a high price for low recall-value. Compared with generative method, discriminative architecture can obtain a higher recall-value with low guarantee for precision-value.

The feature of discriminative architecture indicates that if it determines a candidate hypernym-hyponym relation word pair as negative, then the word pair will have high probability for negative. We can use discriminative architecture to help the generative one to get rid of some false positive instance. For this reason, we fuse the generative and

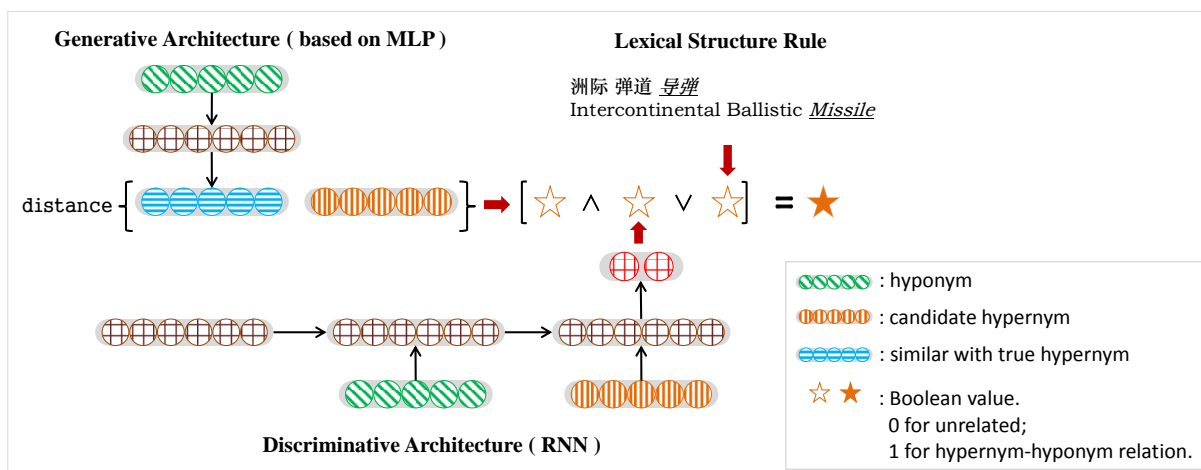

Figure 4: Fusion Learning Architecture.

discriminative architectures together by applying Boolean operator "AND" to the results outputted by the two architectures. Excepting a much higher precision-value than the precious two models and almost the same recall-value as the generative one.

By combining discriminative and generative architectures, the fusion architecture can discovery hypernym-hyponym much more precisely but becomes only focusing deep level semantic and ignoring the lexical stricture information which is very useful for discovery of hypernym-hyponym relationship, especially for compound nouns (CNs), for instance, "洲际弹道导弹(Intercontinental Ballistic Missile)". The root word of CN often indicates a hypernym relation, like the word "导弹(Missile)" is the hypernyms of the precious CN. Root word of a CN can be obtained via using syntax dependency parsing or semantic dependency parsing of CN. Due to the word formation rule of Chinese, the root word is usually the last word in CN segmentation result. To supplement the capacity of learning semantic hierarchy from lexical structure, we use the simple lexical structure rule to assist previous fusion model.

The final fusion learning architecture (showed in Figure 4) is composed of three parts, namely generative architecture, discriminative architecture and lexical structure rule module.

## 4 Experiments

In the experimental stage, we implement our fusion architecture for learning semantic hierarchies. To the end of this, we first introduce the preparation of experimental setup. Next, we report the performance of fusion architecture and its components. Subsequently, we compare the performance of our method to those of several previous methods in different aspects and give an example for construction of semantic hierarchies.

### 4.1 Experimental Setup

**Pre-trained Word Embeddings**

We use a Chinese encyclopedia corpus named Baidubaike[1] to learn word embeddings, which contains about 30 million sentences (about 780 million words). The Chinese segmentation technology is provided by the open-source Chinese language processing platform LTP[2] (Che et al., 2010). Then, we employ the Skip-gram method (Section 3.1) to train word embeddings for the further experiment. We obtain the embedding vectors of 0.56 million words in total.

**Dataset and Evaluation Metrics**

The training data for learning semantic hierarchies is collected from CilinE[3] which contains 100,093 Chinese words and organized as a hierarchy of five levels, in which the words are linked by hypernym–hyponym relations. Finally, we obtain 15,242 word pairs of hypernym–hyponym relation for positive instances and constructed 15,242 negative instances for training.

For comparability we use the same test dataset as Fu et al. (2014) in evaluation stage. They obtain the hypernyms for 418 entities, which are se-

---

[1]Baidubaike (https://baike.baidu.com/) is one of the largest Chinese encyclopedias.
[2]http://www.ltp-cloud.com/demo/
[3]http://www.ltp-cloud.com/download/

| Word Dimension | # of neutrons in hidden layer(RNN) | # of neutrons in hidden layer(MLP) | Batch size | Adadelta parameter |
|---|---|---|---|---|
| $d_w = 300$ | $n_h = 800$ | $n_h = 500$ | $b = 20$ | $\rho = 0.95, \varepsilon = 1e^{-6}$ |

Table 1: Parameters used in our experiments.

lected randomly from Baidubaike, following their previous work (Fu et al., 2013). The final data set was manually labeled and measured the inter-annotator agreement by using the kappa coefficient (Siegel and Castellan, 1981). The kappa value is 0.96, which indicates a good strength of agreement. Training data and test data are showed in Table 2.

| Relation | # of word pairs | |
|---|---|---|
| | Training | Test |
| hypernym-hyponym | 15,242 | 1,079 |
| hyponym-hypernym | 7,621 | 1,079 |
| unrelated | 7,621 | 3,250 |
| Total | 30,484 | 5,408 |

Table 2: The Experimental Data.

We use precision-value, recall-value, and F1-score as metrics to evaluate the performances of the methods. Since the discovery of hypernym-hyponym relation is a binary classification task, we only report the performance of the positive instances recognition in the experiments.

**Parameter Settings and Training**

In our fusion architecture, there are MLP for generation (see in Section 3.2) and RNN for discrimination (see in Section 3.3) need to be trained. We experimentally study the effects of several hyper-parameters on this two neural networks: the number of neutrons in hidden layer, the selection of activation function. Table 1 shows all parameters used in the experiments. We use Adadelta (Zeiler, 2012) in the update procedure, which relies on two main parameters, $\rho$ and $\varepsilon$, which do not significantly affect the performance. Following Zeiler (2012), we choose 0.95 and $1e^{-6}$, respectively, as the values of these two parameters.

In the training stage, we train the discriminative architecture and generative architecture respectively. For training discriminative architecture based on RNN, we use the whole training data. But for training generative architecture based on MLP, we only use the positive instances in train-

ing data as Fu (2014) for generating positive hypernym vectors.

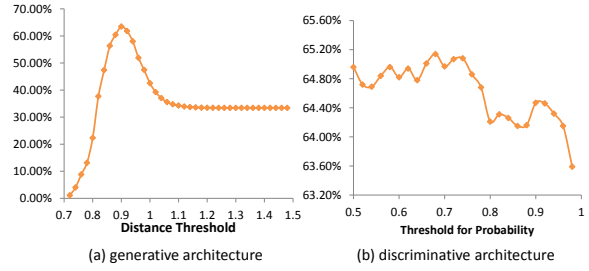

(a) generative architecture  (b) discriminative architecture

Figure 5: The effects on F1-score of threshold for generative and discriminative architecture.

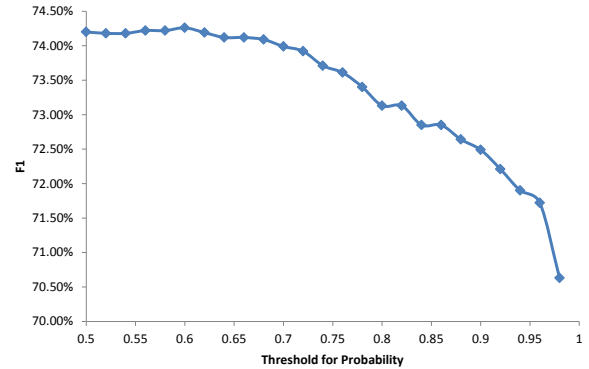

Figure 6: The effects on F1-score of threshold for fusion architecture.

### 4.2 Performance of Fusion Architecture

In the test stage, we tune the performance of architecture by changing the thresholds for components of fusion architecture. There two kinds of thresholds need to be tuned: the similar distance of generative architecture and the probability for positive instances of discriminative architecture.

We first experimentally study the effects of semantic distance threshold on generative architecture based on MLP neural network. Selecting the semantic distance threshold of best F1-score for generative architecture and fixing it, we tun the probability threshold for positive instances of discriminative architecture, observing the F1-score

curve of discriminative architecture and the fusion learning architecture (see in Figure 5, Figure 6).

## 4.3 Comparison with Previous Work

In this section, we compare the proposed method with previous methods, including pairwise hypernym-hyponym relation extraction based on patterns, word distributions, web mining, and based on word embeddings(see in Section 2). Results are shown in Table 3.

| Method | P(%) | R(%) | F1(%) |
|---|---|---|---|
| $M_{Pttern}$ | 97.47 | 21.41 | 35.11 |
| $M_{Snow}$ | 60.88 | 25.67 | 36.11 |
| $M_{balApinc}$ | 54.96 | 53.38 | 54.16 |
| $M_{invCL}$ | 49.63 | 62.84 | 55.46 |
| $M_{Web}$ | 87.40 | 48.19 | 62.13 |
| $M_{Emb}$ | 80.54 | 67.99 | 73.74 |
| $M_{lexicalRule}$ | 100.0 | 16.88 | 28.88 |
| $M_{MLP_{gen.}}$ | 77.96 | 53.51 | 63.46 |
| $M_{RNN_{dis.}}$ | 55.97 | 77.40 | 64.96 |
| $M_{MLP+RNN}$ | 90.00 | 51.48 | 65.50 |
| $M_{Fusion}$ | 91.60 | 62.36 | **74.20** |

Table 3: Comparison of the proposed method with existing methods in the test set.

### Overall Comparison

$M_{Pattern}$ refers to the pattern-based method (Hearst, 1992). The method uses the Chinese Hearst-style patterns (Fu et al., 2013). The result shows that only a small part of the hypernyms can be extracted based on these patterns because only a few hypernym relations are expressed in the fixed patterns, and most of them are expressed in highly flexible manners. $M_{Snow}$ originally proposed by Snow et al. (2004). This method relies on an accurate syntactic parser, and the quality of the automatically extracted patterns is difficult to guarantee.

There are two previous distributional methods $M_{balApinc}$ (Kotlerman et al., 2010) and $M_{invCL}$ (Lenci and Benotto, 2012). Each word is represented as a feature vector in which each dimension is the point-wise mutual information (P-MI) value (Geffet and Dagan, 2005; Zhitomirsky-Geffet and Dagan, 2009) of the word and its context words (see in Section 2).

$M_{Fu}$ refers to a web mining method proposed by Fu et al. (2013) which mines hypernyms of a given word $w$ from multiple sources returning a ranked list of the hypernyms. $M_{Emb}$ refers to a novel method based on word embeddings achieving the best F1-value among previous methods.

$M_{lexicalRule}$ refers to using lexical structure rule to discover hypernym-hyponym relations (see in Section 3.4). The 100% precision-value on test data indicates the lexical rule is correct for most compound nouns (CNs). However the rule only takes effect for CNs which are minority. $M_{MLP_{gen.}}$ represents the generative architecture based on MLP neural network (see in Section 3.2), the method get a higher F1-score than most of previous semantic hierarchy discovery method except $M_{Emb}$. $M_{RNN_{dis.}}$ represents the discriminative architecture based on RNN neural network (see in Section 3.3) which obtains the highest recall-value in comparison of the proposed methods. $M_{MLP+RNN}$ combines these two architectures based on MLP and RNN (see in Section 3.4), getting a much higher precision-value than any components and a comparable recall-value with $M_{MLP_{gen.}}$.

$M_{Fusion}$ refers to a fusion learning architecture composed of discriminative and generative architectures and assisted with lexical structure rule (see in Section 3.4). Assisted with the simple lexical structure rule, the fusion learning architecture get a better F1-score than all of the previous methods do and significantly improve the precision-value over the state-of-the-art method $M_{Emb}$.

Further, we experimentally study the effects of lexical rule for $M_{Emb}$. The results show that the method $M_{Emb+lexicalRule}$ does not improve the F1-score compared with the original $M_{Emb}$. The reason maybe that $M_{Emb}$ already recall the compound nouns (CNs) hypernym-hyponym relations (see in Table 4). The results also indicate that the improvement is mainly caused by discriminative generative architecture.

| | P(%) | R(%) | F1(%) |
|---|---|---|---|
| $M_{Emb+lexicalRule}$ | 80.54 | 67.99 | 73.74 |
| $M_{Fusion}$ | **91.60** | 62.36 | **74.20** |

Table 4: Comparison of the $M_{Emb}$ assisted with lexical structure rule.

### Comparison on the Out-of-CilinE Data

Since the training data is extracted from CilinE, we are greatly interested in the performance of our method on the hypernym-hyponym relations out-

side of CilinE. We assume that as long as there is one word in the pair not existing in CilinE, the word pair is outside of CilinE. In our test data, about 61% word pairs are outside of CilinE.

| | P(%) | R(%) | F1(%) |
|---|---|---|---|
| $M_{Wiki+CilinE}$ | 80.39 | 19.29 | 31.12 |
| $M_{Emb}$ | 65.85 | 44.47 | 53.09 |
| $M_{Fusion}$ | 79.92 | 44.94 | **57.53** |

Table 5: Comparison on the Out-of-CilinE Data.

Table 5 shows the performances of the baseline method $M_{Wiki+CilinE}$, previous state-of-the-art method $M_{Emb}$ and our method $M_{Fusion}$ on the out-of-CilinE data. In comparison, $M_{Wiki+CilinE}$ has the highest precision-value but has a lowest recall-value, $M_{Emb}$ significantly improve recall-value and F1-score. By contrast, our method $M_{Fusion}$ can discover a little bit more hypernym–hyponym relations than $M_{Emb}$ with achieving a more than 14% precision-value improvement. And our method can get an F1-score of 57.53%, which is a new state-of-the-art result on the Out-of-CilinE Data.

**Combined with Manually-Built Hierarchies**

For further exploration, we combine our method $M_{Fusion}$ with the existing manually-built hierarchies in Wikipedia and CilinE. The combination strategy is to simply merge all positive results from the two methods together, and then to infer new relations based on the transitivity of hypernym–hyponym relations. The same manner allied to precious method $M_{Emb}$ to be compared. The comparison is showed in Table 6. Combining our fusion method $M_{Fusion}$ with manually-built hierarchies Wikipedia and CilinE can further improve F1-score to 82.01%, gets an about 1.7% improvement compared with the same manners on $M_{Emb}$.

| | P(%) | R(%) | F1(%) |
|---|---|---|---|
| $M_{Emb}$ | 80.54 | 67.99 | 73.74 |
| $M_{Fusion}$ | 91.60 | 62.36 | **74.20** |
| $M_{Emd+CilinE}$ | 80.59 | 72.42 | 76.29 |
| $M_{Fusion+CilinE}$ | 91.64 | 70.76 | **79.85** |
| $M_{Emd+Wiki+CilinE}$ | 79.78 | 80.81 | 80.29 |
| $M_{Fusion+Wiki+CilinE}$ | 91.00 | 74.63 | **82.01** |

Table 6: Comparison of $M_{Fusion}$ and $M_{Emb}$ combined with manually-built hierarchies.

## 4.4 Example of Learning Semantic Hierarchies

In Figure 7, there is an example of learning semantic hierarchies based on our fusion architecture ($M_{Fusion}$) and combined method using manually-built hierarchies ($M_{Fusion+Wiki+CilinE}$). From the results, we can see that our method can actually learn the semantic hierarchies for a given word and its hypernyms list relatively precisely.

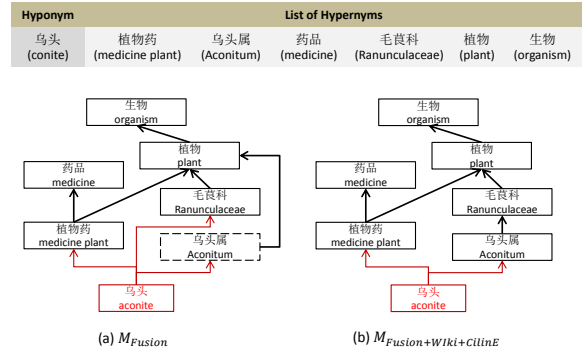

Figure 7: Example of Learning Semantic Hierarchies.

The dashed line frames in Figure 7(a) refers to the losing hypernym-hyponym relations words. For instance, our method fail to learn the semantic hierarchies between "乌头属(aconite)" and "毛茛科(Ranunculaceae)". The reason maybe that their semantic similarity effects representations close to each other in the embedding space and our method can not find suitable projection for these pairs. By combining our method with manually-built hierarchies, we can improve the capacity of learning semantic hierarchies. In this case, the combined method can build the semantic hierarchies correctly (see in Figure 7(b)).

## 5 Conclusion

This paper proposes a novel method for learning semantic hierarchies based on discriminative generative fusion architecture combined with a very simple lexical structure rule. The fusion architecture method can be easily expanded to be suitable for other languages. In experiments, the proposed method achieves the best F1-score of 74.20% on a manually labeled test dataset outperforming state-of-the-art methods with a much higher precision-value of 91.60% for application. Further experiments show that our method is complementary with some manually-built hierarchies to learn semantic hierarchy construction more precisely.

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
