# Peer review of "Constructing Semantic Hierarchies via Fusion Learning Architecture"

_ACL 2017 — decision unknown_

[Official Review · Reviewer 1 · rating 2 · confidence 5]
soundness 5 · originality 3 · clarity 5 · impact 3 · substance 1 · appropriateness 5 · meaningful comparison 4 · presentation format Poster

- Strengths:
- The paper tackles an important issue, that is building ontologies or thesauri
- The methods make sense and seem well chosen
- Methods and setups are well detailed
- It looks like the authors outperform the state-of-the-art approach (but see
below for my concerns)

- Weaknesses:
The main weaknesses for me are evaluation and overall presentation/writing.
- The list of baselines is hard to understand. Some methods are really old and
it doesn't seem justified to show them here (e.g., Mpttern).
- Memb is apparently the previous state-of-the-art, but there is no mention to
any reference.
- While it looks like the method outperforms the previous best performing
approach, the paper is not convincing enough. Especially, on the first dataset,
the difference between the new system and the previous state-of-the-art one is
pretty small.
- The paper seriously lacks proofreading, and could not be published until this
is fixed – for instance, I noted 11 errors in the first column of page 2.
- The CilinE hierarchy is very shallow (5 levels only). However apparently, it
has been used in the past by other authors. I would expect that the deeper the
more difficult it is to branch new hyponym-hypernyms. This can explain the very
high results obtained (even by previous studies)...

- General Discussion:
The approach itself is not really original or novel, but it is applied to a
problem that has not been addressed with deep learning yet. For this reason, I
think this paper is interesting, but there are two main flaws. The first and
easiest to fix is the presentation. There are many errors/typos that need to be
corrected. I started listing them to help, but there are just too many of them.
The second issue is the evaluation, in my opinion. Technically, the
performances are better, but it does not feel convincing as explained above.
What is Memb, is it the method from Shwartz et al 2016, maybe? If not, what
performance did this recent approach have? I think the authors need to
reorganize the evaluation section, in order to properly list the baseline
systems, clearly show the benefit of their approach and where the others fail.
Significance tests  also seem necessary given the slight improvement on one
dataset.

[Official Review · Reviewer 2 · rating 2 · confidence 4]
soundness 5 · originality 3 · clarity 2 · impact 3 · substance 4 · appropriateness 5 · meaningful comparison 4 · presentation format Poster

- Strengths:

  * Knowledge lean, language-independent approach

- Weaknesses:

  * Peculiar task/setting
  * Marginal improvement over W_Emb (Fu et al, 2014)
  * Waste of space
  * Language not always that clear

- General Discussion:

It seems to me that this paper is quite similar to (Fu et al, 2014) and only
adds marginal improvements. It contains quite a lot of redundancy (e.g. related
work in  sec 1 and sec 2), uninformative figures (e.g. Figure 1 vs Figure 2),
not so useful descriptions of MLP and RNN, etc. A short paper might have been a
better fit.

The task looks somewhat idiosyncratic to me. It is only useful if you already
have a method that gives you all and only the hypernyms of a given word. This
seems to presuppose (Fu et al., 2013). 

Figure 4: why are the first two stars connected by conjunction and the last two
starts by disjunction?              Why is the output "1" (dark star) if the the
three
inputs are "0" (white stars)?

Sec 4.2, lines 587-589 appears to suggest that thresholds were tuned on the
test data (?) 

W_Emb is poorly explained (lines 650-652).

Some parts of the text are puzzling. I can't make sense of the section titled
"Combined with Manually-Built Hierarchies". Same for sec 4.4. What do the red
and dashed lines mean?